# Comparing Life Expectancy Determinants between Saudi Arabia and United Arab Emirates from 1980–2020

**Anak Agung Bagus Wirayuda** [1] , **Abdulaziz Al-Mahrezi** [2] **and Moon Fai Chan** [1,*]

1    Department of Family Medicine and Public Health, College of Medicine and Health Sciences,
     Sultan Qaboos University, Muscat 123, Oman; s128499@student.squ.edu.om
2    Director General of Sultan Qaboos University Hospital, Sultan Qaboos University, Muscat 123, Oman;
     abdulaziz@squ.edu.om
*    Correspondence: moonf@squ.edu.om; Tel.: +968-2414-1132

**Abstract:** Despite marked advancements, life expectancy (LE) growth in Saudi Arabia and the United Arab Emirates (UAE) has remained stagnant compared to other developed nations. This study aims to investigate the significant correlation between macroeconomic (ME), sociodemographic (SD), and health status and resources (HSR) factors and LE to formulate an explanatory model for Saudi Arabia and the UAE—a previously unexplored area. Utilizing an ecological, retrospective, time-series study design, we delved into secondary data on SD, ME, and HSR factors and LE of the populations of the UAE and Saudi Arabia spanning three decades (1980–2020). We employed partial least squares–structural equation modeling for statistical analysis. Our analysis revealed significant direct impacts of HSR factors on LE for Saudi Arabia ($\beta = 0.958$, $p < 0.001$) and the UAE ($\beta = 0.716$, $p < 0.001$). Furthermore, we discerned a notable indirect influence of ME factors on LE, mediated through SD and HSR factors for Saudi Arabia ($\beta = 0.507$, $p < 0.001$) and the UAE ($\beta = 0.509$, $p < 0.001$), along with a considerable indirect effect of SD factors on LE through HSR (Saudi: $\beta = 0.529$, $p < 0.001$; UAE: $\beta = 0.711$, $p < 0.001$). This study underscores the mediating role of a nexus of ME–SD–HSR factors on LE in Saudi Arabia and the UAE. Consequently, these findings signal an imperative need for holistic policy interventions addressing ME, SD, and HSR factors, aiming to alter health behaviors and improve LE projections for Saudi Arabia and the UAE in the long run.

**Keywords:** life expectancy; Saudi Arabia; United Arab Emirates; structural equation model; macroeconomic; sociodemographic; health status–resources

## 1. Introduction

Many countries are rappelling with repeated recessions, catalyzing a multifaceted crisis that encapsulates health output in the 21st century [1]. A critical method for assessing the health sphere is to analyze life expectancy (LE) projections, particularly at birth [2,3]. The intriguing subject of "Social Determinants of Health" (SDoH) has been extensively researched within the public health domain [4,5]. Many studies have scrutinized the correlation between SDoH and life expectancy, yielding insightful results. To comprehend the long-term influences on life expectancy, it is imperative to understand the interplay of macroeconomic (ME), sociodemographic (SD), and health status and resources (HSR) factors [6].

Life expectancy projection has always been strongly linked positively to income factors, but some developed countries are still stagnating in their life expectancy improvement [3,7]. According to the World Health Organization (WHO), global life expectancy at birth had increased from 66.8 years in 2000 to 73.4 years in 2019 [7,8]. Meanwhile, according to the OECD iLibrary, life expectancy has increased in all OECD countries over the past 50 years, but progress has slowed over the last decade [9,10]. The COVID-19 pandemic led to life expectancy falling in most OECD countries in 2020 [10]. Other collective nations or groups

need to pay attention to this phenomenon and learn from it, including the Gulf Cooperation Council (GCC).

As part of a network of mainly high-income countries [11], Saudi Arabia and the United Arab Emirates (UAE) are the two major powers representing the GCC region [11,12]. According to the latest WHO data published in 2020, the life expectancy at birth in Saudi Arabia has improved by 3.79 years, from an average of 70.5 years in 2000 to 74.3 years in 2019 [13,14]. On the other hand, life expectancy at birth in the United Arab Emirates (UAE) has improved by 2.9 years, from 73.2 years in 2000 to 76.1 years in 2019 [8]. The total life expectancy at birth in the UAE saw no significant changes in 2021 compared to 2020 and remained at around 78.71 years [15]. Interestingly, the developed and high-income countries worldwide have an average life expectancy of over 81 years [9,16], and none of the GCC members have achieved it yet. This is something that Saudi Arabia and the UAE can strive to achieve to establish themselves as countries with high healthcare performance.

The GCC members' socioeconomic, cultural, and epidemiological attributes exhibit notable parallels [17]. The Gulf region has witnessed unprecedented modernization, defining the socio-political landscape based on the subsequent global order [18,19]. Reliance on the oil industry became a common economic denominator for GCC members due to the substantial economic surge experienced from the 20th to the early 21st century [20,21]. The Gulf Arab ethnicities predominantly inhabit the region, maintaining a cultural homogeneity that cements their shared identity [22]. While similarities in both physical and spiritual lifestyles are evident [23], the expansive migrant worker population in urban areas must not be overlooked [24], as it affirms the comparability of the GCC countries' sociodemographic characteristics, including Saudi Arabia and the UAE [25].

The shared experience of the contemporary epidemiological shift further harmonizes GCC members as they confront similar health challenges [17]. Consequently, the foundation for valuable dialogues concerning the health determinants of these nations can be fortified through contextual comparisons utilizing the life expectancy model. As notably developed GCC members, Saudi Arabia and the UAE enjoyed a steep macroeconomic ascent at the close of the 20th century. Notwithstanding the appreciable strides made through concerted government initiatives, the early 21st-century global financial slowdown, particularly after the plunge and volatility in global oil prices, posed a significant hurdle [26]. Moreover, these nations are aligning with their Gulf counterparts, such as Bahrain, in the transition towards a post-oil era, a significant shift anticipated to be undertaken soon [27].

Regarding macroeconomic considerations, such as foreign exchange rates and monetary policies, it is pertinent to consider broader variables like the Gross Domestic Product (GDP), Gini index, income level, unemployment rate, and inflation rate [28]. A country's economic vitality can improve living conditions, elevating life expectancy [29]. For instance, a robust economy can afford to invest more in healthcare, which may lead to better health outcomes for the population [30]. As one of the world's richest countries in per capita income, the UAE's economy is largely driven by its abundant natural resources, economic diversification, innovation, and the influx of foreign direct investment [31,32]. The country's robust economy has contributed to developing sectors such as tourism and real estate, facilitating the provision of advanced healthcare services and a high standard of living for its citizens [33,34]. Similarly, Saudi Arabia has also seen an improvement in its economy over the years due to various factors such as economic diversification and increased foreign investment. According to the Borgen Project [35], Saudi Arabia is expected to generate more than 60% growth and create jobs in mining and metals, petrochemicals, manufacturing, retail and wholesale trade, tourism and hospitality, finance, construction, and health care [35,36].

Sociodemographic factors, including infant mortality rate, literacy rate, education level, socioeconomic status, population growth, and gender inequality, also significantly shape life expectancy [6,30]. In Saudi Arabia, demographic factors have been reviewed as a significant contributor to life expectancy, and Saudi Arabia resides in the very high human development index category. This situation is expected to provide a basis for

being forward-looking and targeting positive gains in life expectancy [37]. As for the UAE, the population consists primarily of immigrants, with Emirati nationals constituting only about 20% of the total population [38,39]. This multicultural and cosmopolitan society has created a unique social environment that likely influences its residents' health outcomes and life expectancy [38]. However, social challenges such as a wealth gap, high cost of living, obesity, and drug abuse may negatively impact life expectancy [38]. The influence of urbanization and Western culture in the modern era can lead to lifestyle changes for the UAE population, which may impact health outcomes, particularly life expectancy. There is a lack of studies that place a necessary emphasis on sociodemographic determinants in both Saudi Arabia and the UAE.

Health status and resources, such as healthcare facilities, the number of healthcare professionals, public health expenditure, death rates, smoking rates, pollution, and vaccinations, form another crucial determinant of life expectancy [40–45]. The UAE faces environmental threats such as invasive species, carbon footprints, limited water resources, overfishing, waste generation, air pollution, and land degradation [25]. This combination of threats strains the UAE's natural resources and quality of life, which could affect life expectancy. The rise of preventive medicine and the establishment of government programs encouraging healthy lifestyles have notably increased life expectancy in the UAE to the second highest in the WHO Eastern Mediterranean region [46]. On the other hand, Saudi Arabia has the highest healthcare expenditures and percentage of GDP allocated to healthcare among GCC countries [15]. The government has also invested in developing primary health care facilities (PHCCs), the main providers of preventive and curative services at the community level [47]. Unfortunately, despite all the expenses, Saudi Arabia's life expectancy is not at the highest projection among countries in the Middle East [48]. Some environmental issues, e.g., air pollution and water contamination, affect public health in Saudi Arabia. The Saudi government has taken various measures to protect the environment and improve environmental health, such as establishing environmental regulations and legislation, implementing renewable energy projects, rehabilitating natural ecosystems, and conducting environmental research [36,42,43]. Based on a simple observation, both Saudi Arabia and the UAE have lower life expectancy than some high-income countries such as Japan (85.03 years), Switzerland (84.25 years), and Canada (82.96 years) [49]. Hence, a comprehensive understanding of the situation would necessitate more detailed data on healthcare accessibility, community health parameters like nutrition and disease burden, the number of hospitals and medical professionals, healthcare expenditure, mortality rates due to various diseases, vaccination coverage, and pollution level [38].

The future trajectory of life expectancy is far from certain and can be influenced by various drivers of health aspects [50]. Health risks that can be managed through medical care or addressed by broad, population-wide initiatives show the most significant variations between scenarios of current and improved health conditions [50]. This highlights the crucial role of strategies to effectively change modifiable risk factors to reduce early deaths [51]. Life expectancy does not exist in a vacuum; it is intertwined with macroeconomic circumstances, demographic factors, and the availability and quality of health resources [44,45]. Therefore, decision-makers must take a comprehensive approach, considering these elements as interconnected parts of a whole, and design interventions targeting the most impactful health determinants relevant to their specific situation [45,50]. The absence of research in this area, particularly in the Saudi and UAE region, implies a significant gap. Developing a comprehensive model at the population level could offer valuable insights to fill this void, especially for policymakers.

The primary objective of this research is to construct comparative structural models that encapsulate the influences of health status and resources, macroeconomic, and sociodemographic attributes on life expectancy in Saudi Arabia and the UAE. The theoretical model under investigation posits six potential relationships among three latent variables (LVs)—health status and resources, macroeconomic, and sociodemographic—and their impact on life expectancy ($H_1$, $H_2$, $H_3$, $H_4$, $H_5$, and $H_6$). These hypotheses are further

elucidated in the conceptual model (Figure 1). By building upon previous country-specific studies [45,52], this research attempts a multi-comparative analysis at the international level. Such an approach could yield a more comprehensive understanding of the life expectancy model in the context of Gulf Cooperation Council (GCC) countries and offer valuable insights for policy-making.

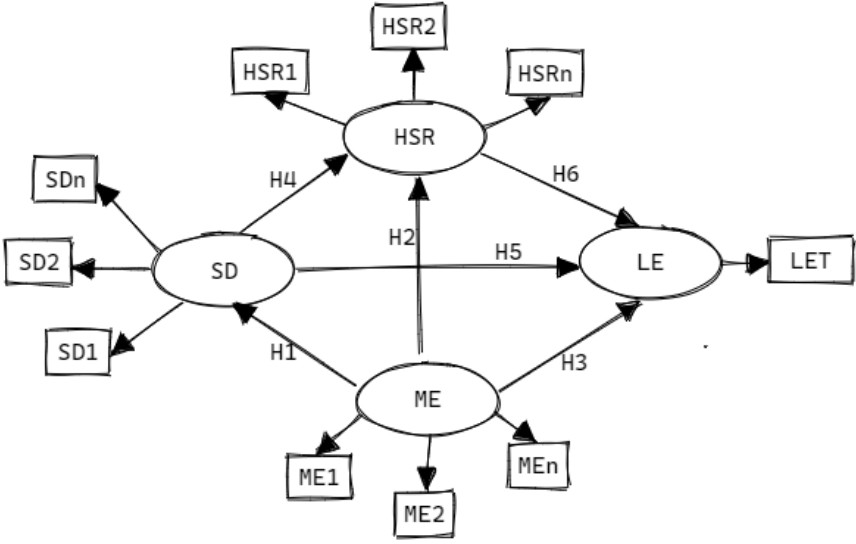

**Figure 1.** The Theoretical Model of the Impact of Macroeconomics, Sociodemographics, and Health Status–Resources on Life Expectancy in Saudi Arabia and the United Arab Emirates (1980–2020). LV: latent variable; MV: manifest variable SD: sociodemographic; ME: macroeconomic; HSR: health status and resources; LE: life expectancy; LET: life expectancy at birth in total (male and female); $H_{1,2,3,4,5,6} \rightarrow$ the direct effect from one LV to the other corresponding LV.

## 2. Materials and Methods

### 2.1. Study Design, Data Results, and Data Collection

This is an ecological retrospective time-series design. The study utilized publicly accessible secondary data from a reputable source, the World Bank database [53,54]. The dataset considers the related macroeconomic, sociodemographic, health status and resources, and life expectancy indicators for citizens of Saudi and UAE nationality. The observation span from 1980 to 2020, as this period has the most data available. Ethical approval exemption was obtained from the Medical Research Ethics Committee of Sultan Qaboos University (MREC #2644).

### 2.2. The Conceptual Model and Statistical Analysis

This research leans towards exploration rather than confirmation purposes, and the required analysis can concurrently handle non-normal data [55] with small sample sizes [56]. So, the Partial Least Squares–Structural Equation Model (PLS-SEM) was employed [55]. To handle missing data, imputation methods in PLS-SEM have not yet been well-developed [57]. We employed a case-wise method, the default method in PLS-SEM, instead of the mean replacement method because the latter method may artificially induce large variance [55,57].

PLS-SEM elucidates the relationships among sociodemographic, macroeconomic, and health status and resources latent variables (LVs) in this study. PLS-SEM delineates the pathways and relationships between LVs and their corresponding manifest variables (MVs). The potency of this study derives from the relational matrix of all variables within the conceptual model (Figure 1). If each LV construct anticipates an $R^2$ minimum of 0.25, the required sample size (years of observation) stands at a minimum of 30 years, adhering to the specific period assessed in this study. Such a sample size can achieve 80% power at a 5% significant level [55,56]. If the data do not support the conceptual and analytical

model, relationship adjustments or exclusions are executed based on initial run results. Data imputation follows the PLS-SEM guideline for missing explanatory variables [56,58]. The data cleaning process utilizes the R-package, and SEM analyses are performed using SmartPLS 4.0®. All analyses adhere to a 5% significance level.

The conceptual model, built on the theoretical framework [45] and preceding systematic review [6], is outlined in Figure 1. Detailed insights into the MVs and LVs are furnished in Table 1. The model encompasses item indicators or MVs, which reflect their associated LVs. Based on the systematic review, all items affect life expectancy significantly. The structural or inner model scrutinizes the relationships between the indicated LVs, as portrayed in Figure 1. The model also provides indirect and total effects, accessible automatically through SmartPLS 4.0®.

**Table 1.** List of Latent and Manifest Variables with Their Respective Loadings.

| LV | MV | Abbreviation | Indicator (Unit) | Loading | |
|---|---|---|---|---|---|
| | | | | Saudi | UAE |
| LE | LET | Life Expectancy in Total | Life expectancy at birth, total (years) | 1.000 * | 1.000 * |
| ME | GDPpc | GDP per capita | GDP per capita (Local Currency Unit) | 0.950 * | 0.961 * |
| | FFE | Fossil Fuel Electricity | Fossil fuels electricity generation (billion kilo-watthours) | 0.966 * | - |
| | | | Electricity production from natural gas sources (%) | - | 0.847 * |
| SD | PPSE | Pre-Primary School Enrollment | School enrollment, pre-primary (% gross) | - | 0.951 * |
| | PSE | Primary School Enrollment | School enrollment, primary (% gross) | - | 0.578 * |
| | SSE | Secondary School Enrollment | School enrollment, secondary (% gross) | - | 0.708 * |
| | TSE | Tertiary School Enrollment | School enrollment, tertiary (% gross) | 1.000 | - |
| HSR | DPTI | Diphtheria, Pertussis, and Tetanus (DPT) Immunization | Immunization, DPT (% of children ages 12–23 months) | 0.972 * | 0.979 * |
| | MI | Measles Immunization | Immunization, measles (% of children ages 12–23 months) | 0.980 * | 0.945 * |
| | TGE | Total Greenhouse Emissions | Total greenhouse gas emissions (kilotons of $CO_2$) | 0.948 * | 0.913 * |

* Significant at $p < 0.05$. LV: latent variable. MV: manifest variable. SD: sociodemographic; ME: macroeconomic; HSR: health status and resources. Source: World Bank [53,54]; UAE: United Arab Emirates; Saudi: Saudi Arabia.

Negative loading MVs in the PLS algorithm are transformed according to PLS-SEM guidelines. The indicator reliability for each MV maintains the loading at a significant value of 5% [59]. In contrast, the internal consistency reliability for each LV takes into account Cronbach's alpha (CA), composite reliability (CR), and rhô-alpha (Rhô-A) at a minimum of 0.7 [59]. Convergent validity for each LV is measured using the average variance extracted (AVE) with a cut-off minimum of 0.50 [59]. Discriminant validity between two reflective constructs (the LVs in this study) is established when the upper bound of the confidence interval (CI) of the HTMT criterion is <1 [60]. $R^2$ and $Q^2$ denote the model's predictability, with an $R^2 > 0.25$ being recommended and a $Q^2 > 0.35$ considered strong [61]. The $f^2$ also measures the direct effect's significance if the result is >0.35 [60]. A radar chart is utilized for comparative visualization of the results, displaying the visual comparison of the total effects from both countries ($H_1$ to $H_6$).

## 3. Results

The final model of the study presents a varied number of MVs for the sociodemographic, macroeconomic, and health status and resources LVs in both Saudi Arabia and the UAE, as well as for the life expectancy in the two models. Table 1 and Figures 2 and 3 display each MV, showing significant loadings for each model.

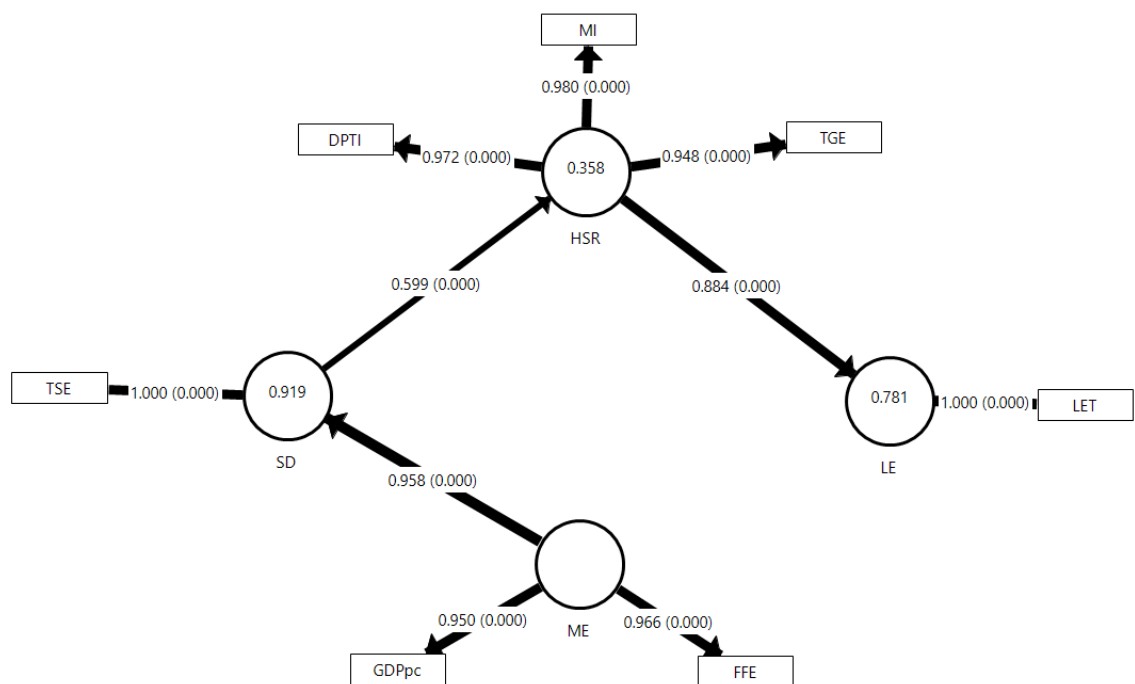

**Figure 2.** The Final Model of the Impact of Macroeconomics, Sociodemographics, and Health Status and Resources on Life Expectancy in Saudi Arabia (1980–2020). SD: sociodemographic; ME: macroeconomic; HSR: health status–resources; LE: life expectancy; LET: Average life expectancy at birth (both males and females) in Saudi Arabia; GDPpc: gross national income per capita; FFE: fossil fuel electricity; TSE: tertiary school enrollment; DPTI: DPT immunization; MI: measles immunization; TGE: total greenhouse emissions; numbers inside the LV circle indicate the $R^2$ value. All the values inside the diagram above are significant at $p < 0.05$.

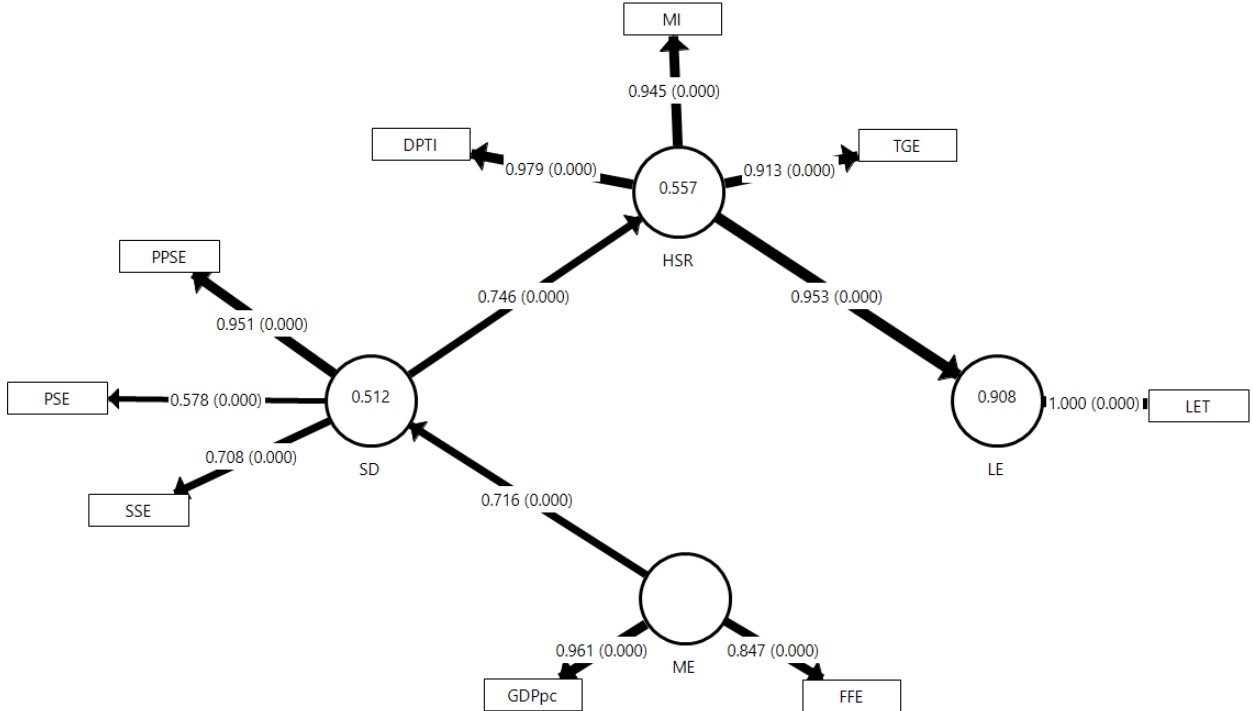

**Figure 3.** The Final Model of the Impact of Macroeconomics, Sociodemographics, and Health Status and Resources on Life Expectancy in the United Arab Emirates (1980–2020). SD: sociodemographic;

ME: macroeconomic; HSR: health status and resources; LE: life expectancy; LET: life expectancy in total (both males and females) in UAE; GDPpc: gross domestic product per capita; FFE: fossil fuel electricity; PPSE: pre-primary school enrollment; PSE: primary school enrollment; SSE: secondary school enrollment; DPTI: DPT immunization; MI: measles immunization; TGE: total greenhouse emissions; numbers inside the LV circle indicate the $R^2$ value. All the values inside the diagram above are significant at $p < 0.05$.

Table 2 shows the final models' reliability, validity, and predictability for each LV. All LVs demonstrate a Cronbach's alpha (CA), composite reliability (CR), Rhô-Alpha (Rhô-A), and average variance extracted (AVE) that meet or exceed their respective thresholds. In the Saudi model, macroeconomic measures are as follows: CA = 0.930, CR = 0.952, Rhô-A = 0.957, and AVE = 0.775. For the UAE model, macroeconomic metrics are CA = 0.800, CR = 0.901, Rhô-A = 0.914, and AVE = 0.734. Concerning the sociodemographic metrics, the Saudi model demonstrates a score of 1.000 across CA, CR, Rhô-A, and AVE, while for UAE, these values are 0.644, 0.799, 0.953, and 0.523, respectively. The health status and resources parameter for Saudi is CA = 0.965, CR = 0.977, Rhô-A = 0.972, and AVE = 0.934, and for UAE, CA = 0.952, CR = 0.965, Rhô-A = 0.943, and AVE = 0.843. The life expectancy metric for both nations scores 1.0 in all areas.

**Table 2.** Reliability, Validity, and Predictability of the Latent Variables.

| LV | Country | CA | CR | Rhô-A | AVE | $Q^2$ | $R^2$ | HTMT (95%CI) | | |
|----|---------|-----|-----|-------|-----|-------|-------|--------------|---|---|
| | | | | | | | | ME | SD | HSR |
| ME | Saudi | 0.930 | 0.952 | 0.957 | 0.775 | - | - | - | - | - |
| | UAE | 0.800 | 0.901 | 0.914 | 0.734 | - | - | | | |
| SD | Saudi | 1.000 | 1.000 | 1.000 | 1.000 | 0.590 | 0.919 | 0.668 (0.258–0.983) | - | - |
| | UAE | 0.644 | 0.799 | 0.953 | 0.523 | 0.740 | 0.512 | 0.783 (0.573–0.960) | - | |
| HSR | Saudi | 0.965 | 0.977 | 0.972 | 0.934 | 0.640 | 0.358 | 0.800 (0.702–0.988) | 0.524 (0.274–0.803) | - |
| | UAE | 0.952 | 0.965 | 0.943 | 0.843 | 0.720 | 0.557 | 0.847 (0.723–0.955) | 0.822 (0.633–0.981) | |
| LE | Saudi | 1.000 | 1.000 | 1.000 | 1.000 | 0.864 | 0.781 | 0.921 (0.821–0.975) | 0.656 (0.293–0.913) | 0.898 (0.852–0.964) |
| | UAE | 1.000 | 1.000 | 1.000 | 1.000 | 0.969 | 0.908 | 0.933 (0.849–0.993) | 0.978 (0.945–0.998) | 0.972 (0.946–0.990) |

CA: Cronbach's alpha (>0.7); Rhô -A: rhô-alpha (>0.7); CR: composite reliability (>0.7); AVE: Average variance extraction (>0.50); $R^2$: >0.25 recommended; $Q^2$: <0.02 (weak), 0.15 (moderate), >0.35 (strong); SD: sociodemographic; ME: macroeconomic; HSR: health status and resources; LE: life expectancy; LV: latent variables; HTMT: hetero-trait and mono-trait ratio (95% CI < 1.0); all estimates are significant at $p < 0.05$; UAE: United Arab Emirates; Saudi: Saudi Arabia.

The models' $Q^2$ value indicates strong predictive relevance for each LV in both countries, with $R^2$ values explaining a substantial percentage of variance in each LV for both Saudi and UAE models. The $Q^2$ value of the Saudi model spans 0.590 for sociodemographic, 0.640 for health status–resources, and 0.864 for life expectancy. In contrast, for UAE, it spans 0.740 for sociodemographic, 0.720 for health status–resources, and 0.969 for life expectancy. Saudi's model elucidates 91.9% of the sociodemographic variance, 35.8% of health status–resources, and 78.1% of life expectancy. In comparison, the UAE model explicates 51.2% of sociodemographic, 55.7% of health status and resources, and 90.8% of life expectancy variances.

The HTMT criterion for each relationship in the final models falls below 1.0, confirming discriminant validity. Details of these values are in Table 2 for each model and LV. The HTMT for the Saudi model ranges from 0.524 to 0.921, while for the UAE model, it ranges from 0.783 to 0.978.

Table 3 summarizes all information regarding direct, indirect, and total effects, and each effect demonstrates significance as the $f^2$ value of each provided direct effect exceeds 0.35. Figure 4 visually portrays the general differences in total effects between the two countries according to the hypothesized relationships ($H_1$ to $H_6$) via a radar chart. Regarding the Saudi model, the $H_1$: ME→SD, $H_2$: ME→HSR, and $H_3$: ME→LE hypotheses have direct and indirect effects ranging from 0.507 to 0.958 ($p < 0.001$), and the $f^2$ ranging from

11.293 for $H_1$ to 0.529 for $H_5$. The $H_4$: SD→HSR and $H_6$: HSR→LE hypotheses indicate direct effects of 0.599 and 0.884, respectively, with $f^2$ values of 0.559 and 3.558, respectively. For the UAE model, the $H_1$: ME→SD, $H_2$: ME→HSR, and $H_3$: ME→LE hypotheses yield direct and indirect effects ranging from 0.509 to 0.716 ($p < 0.001$), with $f^2$ values of 1.051 for $H_1$ and 0.711 for $H_5$. The $H_4$: SD→HSR and $H_6$: HSR→LE hypotheses indicate direct effects of 0.746 and 0.953, respectively, and $f^2$ values of 1.256 and 9.870, respectively.

**Table 3.** Effect Size, Direct Effect, Indirect Effect, and Total Effect of the Latent Variables.

| Hypothesis (Relationship) | Country | Direct Effect (95% CI) | Indirect Effect (95% CI) | Total Effect (95%CI) | $f^2$ |
|---|---|---|---|---|---|
| $H_1$ (ME→SD) | Saudi | 0.958 (0.943–0.972) | - | 0.958 (0.943–0.972) | 11.293 |
| | UAE | 0.716 (0.625–0.779) | | 0.716 (0.625–0.779) | 1.051 |
| $H_2$ (ME→HSR) | Saudi | - | 0.574 (0.495–0.648) | 0.574 (0.495–0.648) | - |
| | UAE | - | 0.534 (0.434–0.603) | 0.534 (0.434–0.603) | - |
| $H_3$ (ME→LE) | Saudi | - | 0.507 (0.415–0.590) | 0.507 (0.415–0.590) | - |
| | UAE | - | 0.509 (0.411–0.576) | 0.509 (0.411–0.576) | - |
| $H_4$ (SD→HSR) | Saudi | 0.599 (0.517–0.674) | - | 0.599 (0.517–0.674) | 0.559 |
| | UAE | 0.746 (0.627–0.823) | | 0.746 (0.627–0.823) | 1.256 |
| $H_5$ (SD→LE) | Saudi | - | 0.529 (0.432–0.614) | 0.529 (0.432–0.614) | - |
| | UAE | - | 0.711 (0.590–0.793) | 0.711 (0.590–0.793) | - |
| $H_6$ (HSR→LE) | Saudi | 0.884 (0.815–0.923) | - | 0.884 (0.815–0.923) | 3.558 |
| | UAE | 0.953 (0.928–0.970) | | 0.953 (0.928–0.970) | 9.870 |

LV: latent variable; SD: sociodemographic; ME: macroeconomic; HSR: health status and resources; LE: life expectancy; $f^2$: effect size <0.02 (weak), 0.15 (moderate), >0.35 (strong); all effects are significant at $p < 0.05$. UAE: United Arab Emirates; Saudi: Saudi Arabia.

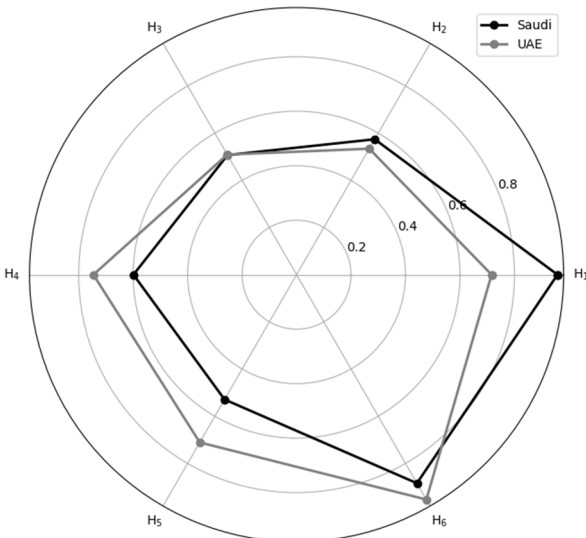

**Figure 4.** Radar Chart for the Comparison of the Total Effects between Saudi Arabia and United Arab Emirates (UAE). The chart is drawn based on the hypothesis relationship ($H_1$ to $H_6$).

## 4. Discussion

### 4.1. Comparison between the Two Models

All the models' performances are satisfactory according to the PLS-SEM principle. For the Saudi model, all hypotheses ($H_1$ to $H_6$) show significant direct or indirect effects, with $f^2$ values indicating a strong impact. The path macroeconomic to sociodemographic ($H_1$) displays the strongest effect, while the path health status–resources to life expectancy ($H_6$) shows the strongest effect on life expectancy as the endpoint. For the UAE model, similar significant direct or indirect effects are found for all hypotheses ($H_1$ to $H_6$), with the path macroeconomic to sociodemographic ($H_1$) demonstrating a significant direct effect. The

path of health status and resources to life expectancy ($H_6$) shows the strongest direct effect, echoing the findings in the Saudi Arabia model.

Contemplating the major pattern of the two final models (Figures 2 and 3), as is also shown in the radar chart (Figure 4), we can see the nexus of ME→SD→HSR→LE as the mediating pattern of the effect within the model. This mediating pattern can provide wisdom by revealing the causal processes that underlie health disparities and by suggesting potential interventions that target the mediators to improve health outcomes [62], which is shown by comparing $H_3$, $H_5$, and $H_6$ altogether, as these are the main relationships that affect life expectancy in this study. We can figure out that both countries have not utilized their sociodemographics and macroeconomics optimally to affect life expectancy. At the same time, they exemplify the direct effect of health status and resources on life expectancy more substantially. We can see this by comparing how macroeconomic and sociodemographic factors impact health status and resources ($H_2$ and $H_4$), which are not as strong as macroeconomic factors affecting sociodemographic ones directly ($H_1$), especially in the Saudi case. This may imply that the sociodemographic and macroeconomic determinants can still be improved to integrate the effect well within the complete model. This might be one of the reasons why Saudi Arabia and the UAE are still relatively slow in their life expectancy projection compared to other high-income countries, despite all their healthcare expenditures and advancements [15,49].

Each model incorporates two macroeconomic indicators: gross domestic product per capita (GDPpc) and fossil fuel electricity production (FFE), all of which have been found to influence macroeconomic factors and life expectancy significantly. GDPpc is an important economic development indicator positively correlated with life expectancy [30,63,64]. This is also obvious from previous studies in Oman [52] and Bahrain [45]. GDP per capita can show how economic growth performs while considering population growth, which is more reliable in terms of inter-country comparisons. Electricity production, which utilizes UAE fossil fuel to a certain degree, is also positively correlated with life expectancy as it indicates a country's infrastructure and technological development [45,65,66]. It is also notable that the Saudi and UAE efforts to diversify their economy away from oil could have important implications for life expectancy by potentially leading to broader economic development, improved living standards, and increased investments in social sectors such as health and education [33,67]. The sole total effect of macroeconomic factors is more substantial than the other key determinants, such as sociodemographic factors and health status–resources. Also, the results indicate that healthcare directly impacts life expectancy and is significantly affected by macroeconomic and sociodemographic factors. The model suggests that while economic growth can initially elevate life expectancy, further growth is fostered by sociocultural and healthcare aspects, which was also noted by other literature [68–71].

For sociodemographic determinants, education levels, such as pre-primary education (PPSE), primary education (PSE), secondary education (SSE), and tertiary education (TSE), within a population had a positive impact on life expectancy. One explanation may be that people with more education earn higher wages, enabling them to purchase proper nutrition and common illness-preventative measures [4]. This finding might suggest that people with more education are more aware of the importance of obtaining better healthcare services to improve their health [5]. Thus, it may be implied that a country's life expectancy will increase as the population's education levels increase [4–6].

For health status and resource determinants, vaccination can classically be referred to as the most important factor that boosted life expectancy in the last century, along with nutrition and antibiotic invention [72]. DPT immunization (DPTI) and measles immunization (MI) are significant aspects of the complete model. Total greenhouse emissions (TGE) can be seen as one environmental indicator within the complete model [40]. TGE may act as a reminder to reduce carbon emissions to improve human life expectancy, even globally [40].

The study emphasizes the importance of considering a broad range of factors when developing strategies to improve health outcomes and extend life expectancy. The re-

sults suggest that focusing solely on disease and morbidity, without considering broader economic and sociodemographic variables, can impair health outcomes in the long run. Cultivating health literacy and sociocultural tactics to foster healthier habits is paramount, particularly in the face of the ubiquity of obesity and other non-communicable diseases in regions like Saudi Arabia and the UAE. Assimilating these aspects into public health strategies can help ensure a comprehensive and effective approach to improving health outcomes at a population level.

However, it is crucial to note that while this model provides valuable insights, it is inherently observational, and the inference from the final model should consider how each LV can affect the others. Also, the relevance of certain factors might change over time due to changes in the population or how variables are measured. Therefore, the model's applicability may need to be reassessed periodically to ensure its continued relevance.

Finally, while the study provides an in-depth analysis of the factors influencing life expectancy in Saudi Arabia and the UAE, it is important to note that these findings may not be generalizable to other contexts, as life expectancy determinants can vary widely between different countries and regions due to differences in socioeconomic conditions, healthcare systems, cultural factors, and other variables.

### 4.2. Limitations of This Study

This study's findings are based on an ecological and longitudinal approach with inherent limitations. Over long periods, changes in the population or how variables are measured could impact the results, and the importance of certain factors might shift over time. The inferences drawn from the final model should be tempered by the understanding that they are observational and shed light on potential causal relationships between the latent variables rather than definitively establishing causality [73].

### 5. Conclusions

The results of this analysis underscore the interplay of macroeconomics, sociodemographics, and health status and resources in shaping a population's health and behavior. As such, improvements and interventions must encompass macroeconomic conditions and sociodemographic and health status–resources factors. Focusing solely on disease and morbidity while neglecting broader economic and sociodemographic variables could detrimentally affect health outcomes in the long run. The importance of fostering health literacy and sociocultural strategies to promote healthier habits cannot be overstated, especially given the prevalence of both communicable and non-communicable diseases in the regions. By incorporating these elements into public health strategies, we can develop a more holistic and effective approach to improving health outcomes at the population level. This comprehensive strategy should be flexible and adaptable to changes in the drivers of health, as these drivers may shift over time and under different circumstances. Thus, ongoing monitoring and evaluation of health determinants are essential for continually refining public health policies and interventions.

**Author Contributions:** Conceptualization, M.F.C.; methodology, M.F.C. and A.A.B.W.; formal analysis, A.A.B.W.; data curation, M.F.C. and A.A.B.W.; writing-original draft preparation: A.A.B.W. and M.F.C.; writing—review and editing, M.F.C., A.A.B.W. and A.A.-M.; supervision, M.F.C. and A.A.-M.; funding acquisition, M.F.C. and A.A.-M. All authors have read and agreed to the published version of the manuscript.

**Funding:** This research was funded by the International Research Collaboration Co-Fund QU-SQU: CL/SQU-QU/MED/22/01.

**Institutional Review Board Statement:** Ethic Committee Name: The Medical Research Ethics Committee, College of Medicine and Health Sciences, Sultan Qaboos University; Approval Code: MREC#2644; Approval Date: 8 November 2021.

**Informed Consent Statement:** Not applicable.

**Data Availability Statement:** The data used in this study can be found online.

**Acknowledgments:** The authors would like to acknowledge the comments from the reviewers.

**Conflicts of Interest:** The authors declare no conflict of interest.

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
