# Peer review of "Comparing Life Expectancy Determinants between Saudi Arabia and United Arab Emirates from 1980–2020"

_ejihpe, doi:10.3390/ejihpe13070095_

Round 1

Reviewer 1 Report

Dear author(s), 

Your manuscript covers a very interesting topic and it was a real pleasure to read it. No further comments

Author Response

Reviewer comments: Your manuscript covers a very interesting topic and it was a real pleasure to read it. No further comments.

Response: Thank you for your great support.

Reviewer 2 Report

I found the work/paper interesting, well-prepared, and designed. I understand the limits of such work and the advantages in using some datasets, but a possible integration is one or two specific studies - even using "simple" questionnaires in one or two local areas. It is clear that this study confirmed that there are no shortcuts to LE, the model is still valid, and this work confirms it - in order to improve the health of the population you need to work on the positive determinants for health.

The English, is clear and not a problem at all.

Author Response

Reviewer comments: I found the work/paper interesting, well-prepared, and designed. I understand the limits of such work and the advantages in using some datasets, but a possible integration is one or two specific studies - even using "simple" questionnaires in one or two local areas. It is clear that this study confirmed that there are no shortcuts to LE, the model is still valid, and this work confirms it - in order to improve the health of the population you need to work on the positive determinants for health.

Response: Thank you for your comments and support. Our team will continue more research on the positive health determinants as suggested.

Reviewer 3 Report

This is an interesting study that provides insight into the factors associated shifting life expectancies in two Gulf states. There has been long-standing debate regarding macro-level variables that influence core demographic indicators such as life expectancy and fertility. While this article doesn’t settle this debate, it provides further evidence that at the national level, macro-level shifts can result in positive changes in human health.

I have some specific comments for the authors to address:

I find the use of the acronyms ME, SD, HSR, and LE to be cumbersome and make the article less readable. Consider using plain language instead.

The first sentence of the article (line 29) is sensationalistic and unnecessary for the paper.

The reference [5] is incorrect and needs to be corrected. Indeed, there are many other references on the SDoH that might be preferable than one specific to Canada - which is not under study in this paper.

The second paragraph should be split at line 46.

The sentence at lines 78-79 is redundant with previous paragraphs.

It is unclear how the paragraph (lines 72-84) justify the conclusion in lines 84-88. I’m not saying it’s untrue, it’s that the paragraph doesn’t lead to this conclusion. Why does a shift to a post-oil era mean that governments need to understand life expectancy models? 

The details on life expectancy modelling (lines 89-91) in the subsequent paragraph is a self-citation [6] - an additional source should be included here. Additionally, reference [29] doesn’t appear to relate to this statement and is a website with data tables.

The introductory section is rather long and could be significantly shortened without losing information.

Many of the included citations are from newspapers or websites rather than academic sources. These should be reduced or replaced.

The captions for Figures 2 & 3 are incorrectly formatted.

The performance of the models given significant missing data should be discussed in more detail (lines 190-196). The justification for this model is, again, a self-citation [52], and should be further justified.

The paragraph from lines 337-340 is unclear and should be re-written.

There should be more care taken in editing. For example, the sentence in line 286 is incomplete with incorrect subject-verb agreement.

This is an interesting study that provides insight into the factors associated shifting life expectancies in two Gulf states. There has been long-standing debate regarding macro-level variables that influence core demographic indicators such as life expectancy and fertility. While this article doesn’t settle this debate, it provides further evidence that at the national level, macro-level shifts can result in positive changes in human health.

I have some specific comments for the authors to address:

I find the use of the acronyms ME, SD, HSR, and LE to be cumbersome and make the article less readable. Consider using plain language instead.

The first sentence of the article (line 29) is sensationalistic and unnecessary for the paper.

The reference [5] is incorrect and needs to be corrected. Indeed, there are many other references on the SDoH that might be preferable than one specific to Canada - which is not under study in this paper.

The second paragraph should be split at line 46.

The sentence at lines 78-79 is redundant with previous paragraphs.

It is unclear how the paragraph (lines 72-84) justify the conclusion in lines 84-88. I’m not saying it’s untrue, it’s that the paragraph doesn’t lead to this conclusion. Why does a shift to a post-oil era mean that governments need to understand life expectancy models? 

The details on life expectancy modelling (lines 89-91) in the subsequent paragraph is a self-citation [6] - an additional source should be included here. Additionally, reference [29] doesn’t appear to relate to this statement and is a website with data tables.

There are a significant number of popular media and websites included as citations. These should be reduced or replaced with peer-reviewed academic sources.

The introductory section is rather long and could be significantly shortened without losing information.

The captions for Figures 2 & 3 are incorrectly formatted.

The performance of the models given significant missing data should be discussed in more detail (lines 190-196). The justification for this model is, again, a self-citation [52], and should be further justified.

The paragraph from lines 337-340 is unclear and should be re-written.

There should be more care taken in editing. For example, the sentence in line 286 is incomplete with incorrect subject-verb agreement.
